# Quantitative Assessment of Water Quality Improvement by Reducing External Loadings at Lake Erhai, Southwest China

**DOI:** 10.3390/ijerph20065038

**Published:** 2023-03-13

**Authors:** Falu Gong, Liancong Luo, Huiyun Li, Lan Chen, Rufeng Zhang, Guizhu Wu, Jian Zhang, Weiqiang Shi, Fan Zhang, Hao Zhang, Ting Sun

**Affiliations:** 1Institute for Ecological Research and Pollution Control of Plateau Lakes, School of Ecology and Environmental Science, Yunnan University, Kunming 650504, China; 2Nanjing Institute of Geography and Limnology, Chinese Academy of Sciences, Nanjing 210008, China

**Keywords:** external loadings, water quality, DYRESM–CAEDYM, Lake Erhai

## Abstract

To quantitatively evaluate the effects on water quality improvement caused by reducing external loadings entering Lake Erhai through inflow rivers, a one-dimensional hydrodynamic and ecological model (DYRESM–CAEDYM) was set up to simulate the water quality and water level variations. The calibrated and validated model was used to conduct six scenarios for evaluating the water quality responses to different amounts of external loading reduction at Lake Erhai. The results show (1) the total nitrogen (TN) concentration of Lake Erhai will be higher than 0.5 mg/L without any watershed pollution control during April–November 2025, which cannot meet Grade II standard of the China Surface Water Environmental Quality Standards (GB3838-2002). (2) External loading reductions can significantly reduce the concentrations of nutrients and Chla at Lake Erhai. The effects of water quality improvement will be proportional to the reduction rate of external loading reductions. (3) Internal release might be an important source of pollution It needs to be seriously considered as well as external loading for mitigating the eutrophication at Lake Erhai in the future.

## 1. Introduction

Under the influence of climatic changes and anthropogenic activities, eutrophication, caused by the excess input of nitrogen and phosphorus, has become a widespread environmental problem in Chinese lakes and reservoirs, which can not only lead to water quality deterioration, bloom outbreaks, biodiversity reduction, and ecosystem degradation, but can also cause great harm to human production and life [1]. Sources of nitrogen and phosphorus can be divided into riverine inputs by inflows (external loads) and release or resuspended from sediment (internal loads). Internal loads include the resuspended nutrients from sediment caused by external disturbances in shallow lakes, and the released nutrients from bottom waters with anaerobic environment in deep-water lakes [2,3]. Excessive inputs of external loads to lakes cannot be self-digested and recovered, causing water quality deterioration and eutrophication for most lakes [4]. For example, the annual mean concentrations of total nitrogen (TN), total phosphorus (TP), and chlorophyll a (Chla) at Lake Poyang significantly increased from the 1980s to 2018, which suggests that there was obviously an increase of external pollutions [5]. At Lake Dianchi, the external loads entreing lake water through inflow rivers accounted for 65.7% of the total pollution inputs from catchment in 2018 [6]. External loads have significantly contributed to eutrophication of Lake Wuli, which is the north part of Lake Taihu, Jiangsu Province. The water quality at the wet season is normally worse than that of the dry season due to rainfall, which can bring considerable pollutants to the lake water through inflow rivers at Lake Wuli [7]. Therefore, reducing external loads is necessary for improving lake water quality and restore lake ecosystem.

The methods to find the impacts of external pollution loadings on water quality include water quality monitoring, field experiments, and model simulations [8,9]. Conventional water quality monitoring methods can accurately measure the concentration of pollutants, but they are time-consuming and expensive. Additionally, it is difficult for the conventional methods to achieve the objective of continuous monitoring and short-term warning of lake water quality. In contrast, modeling can adequately reflect the temporal and spatial variation of lake water quality, and it can be used to analyze the variation in lake water quality under different external loadings reduction rates through scenario simulation, providing an important theoretical basis for governments to scientifically plan and manage lake environment [10,11]. Because of these characteristics, model approach has become a practical and reliable method for intelligent management at many lakes. For example, Li et al. used the Xin′anjiang model and real time high-frequency monitoring data of TP to calculate the TP pollution loads caused by rainstorms. The TP loadings accounted for 69.4% of the total inputted TP loads from catchment for May 2020–April 2021 at Lake Qiandao [12]. Zhao et al. concluded that the water quality of Lake Yilong could not meet its water quality target (Grade III) if both of the reduction rate of external TN and TP loads are less than 77% based on EFDC simulations [10]. DYRESM–CAEDYM is a one-dimensional hydrodynamic and ecological model. It has been widely used in both domestic and oversea freshwater lakes including Lake Taihu [13] and Lake Chaohu [14] in China, Lake Ravn in Denmark [15] and Lake Rotorua in New Zealand [16]. In this paper, DYRESM–CAEDYM was used to detect the water quality responses to external loading reductions at Lake Erhai.

Lake Erhai, the second largest freshwater lake in Yunnan Province, is an important source of drinking water for Dali City. According to water-quality monitoring data during 2000–2020, the water quality has met Grade II of GB3838-2002 for a long time (Table 1). The water quality gradually degraded with eutrophication state changing from oligotroph to mesotrophic during 1988–2013 [17]. There have been many reports about the perspective of pollutant sources, spatial and temporal distribution of nitrogen and phosphoru [18], bloom risk assessment [19], and water environment problem diagnosis [20] in this lake. Although many ecological and environmental engineering projects have been conducted at Lake Erhai, the water quality can only meet Grade III and occasionally meet Grade II according to GB3838-2002 through the year [21]. According to the recently issued *14th Five-Year Plan for Water Pollution Control of Lake Erhai Watershed* and the *Action Plan for the Protection and Governance of Lake Erhai* in Yunnan Province issued in 2022, over CNY 11.7 billion is expected to be invested in the control of lake eutrophication during 2021–2025. It is expected that the water quality of 27 main inflow rivers meet Grade Ⅲ or Grade II, and the water quality at all the lake monitoring sites will meet Grade Ⅱ by 2025.

The objective of the present study was to quantitatively evaluate the improvement of water quality at Lake Erhai, under different reduction rates of external loading. We set up six scenarios with reduction rates of 10% (S1), 20% (S2), 30% (S3), 40% (S4), and 50% (S5), and the water quality of inflow rivers meets Grade III (S6). Then we evaluated the water quality in 2025 under different scenarios comparing to that in 2020 (baseline).

## 2. Study Site

Lake Erhai (25°36′~25°58′ N, 100°06~100°18′ E) is a subtropical plateau lake located at Dali City of Yunnan Province, Southwest China. It is a typical inland faulted freshwater lake with a total water surface area of 252 km^2^ and a total volume of 3 billion m^3^ when the elevation is 1966 m (i.e., the Yellow Sea’s elevation). Its average depth is 10.8 m, with a maximum depth of 21.3 m [21,22]. The basin, with an area of 2565 km^2^, has a subtropical plateau monsoon climate, with an annual average temperature of 15.1℃ and an annual average rainfall and evaporation of 1060 mm and 1208 mm, respectively [23]. There are 117 rivers and streams entering Lake Erhai. The three northern rivers (Miju River, Luoshi River, and Yong’an River), the 18 streams from Cang Mountain in the west, and the Boluo River in the south are the main inflows (Figure 1), accounting for more than 80% of the annual average water supply of the lake [24]. The main natural output river is the Xi’er River, located at the southern end, with an average annual outflow of 697 million m^3^. There is a water diversion channel (Er-Bin Channel) which was constructed in 1994. Its average annual outflow is 71 million m^3^ [20]. The population is about one million living in 167 villages and 17 towns at the catchment. More than 70% of the population are working as agricultural farmers [25]. The agricultural land and population are mostly distributed along the catchment rivers and other lakes located at the north and west areas of Lake Erhai. The pollutants produced by agricultural farming and domestic sewage have been threatening the lake environment for the past three decades.

## 3. Materials and Methods

### 3.1. Methods

#### 3.1.1. DYRESM–CAEDYM

DYRESM (Dynamics Reservoir Simulation Model) and CAEDYM (Computational Aquatic Ecosystem Dynamics Model) is a one-dimensional coupled hydrodynamic–ecological model, developed by the Water Research Center of the University of Western Australia.

DYRESM can simulate and predict the vertical distribution of temperature, salinity, and density in lakes and reservoirs, satisfying the one-dimensional approximation. The model provides a means of predicting the seasonal and interannual variability of lakes and reservoirs, as well as sensitivity testing for long-term changes in environmental factors or watershed properties [26]. The files contained in the model include physical parameters (.par), configuration (.cfg), inflow (.inf), outflow (.wdr), and weather (.met). Both outflow and weather files contain daily data. Listed the physical parameters in Table 2. This file also includes the time of day for output, bubbler entrainment coefficient, and buoyant plume entrainment coefficient, for which we have not changed the values.

(1) DYRESM is based on an assumption of one-dimensionality so that the variations in horizontals direction can be ignored in lakes. Thus, the number and spatial location of rivers have no effect on the results of the model. The inflow file includes daily inflow volumes, water temperature (WT), and water quality information, while the outflow file includes daily outflow volumes. Due to the large number of rivers and streams of Lake Erhai, it is difficult to obtain all the monitoring data for each river from 2000 to 2020. Therefore, the 117 inflow rivers/streams were merged and regarded as one inflow river. The outflow river (Xi’er River) and the diversion channel (Er-Bin Channel) were merged into one outflow river. The total annual amounts of inflows and outflows for 2000–2020 were obtained from the local management authority. Given that the total inflows are linearly proportional to rainfall, the total monthly inflows for the merged inflow river of model can be calculated based on the monthly rainfalls. Then the total inflow in a month was evenly distributed to all the days in this month to produce daily inflows. The daily outflows were calculated with the same method. The estimated daily inflows and outflows were then adjusted based on measured water levels (WL) and water balancing. The WT and quality data used in the inflow files were from monthly measurements.

(2) Meteorology is an important driving factor in determining lake water stability. DYRESM–CAEDYM requires daily meteorological data as inputs, including air temperature (℃), wind speed (m/s), rainfall (m), shortwave radiation (W/m^2^), and vapor pressure (hPa). Meteorological input data were from Dali Meteorological Bureau for 2000–2020. The vapor pressure was calculated by the average relative humidity and air temperature, using the following formula:(1)$$e_{a}=\left(h/100\right)exp\left[2.303\left(\left(7.5q_{D}/\left(q_{D}+237.3\right)\right)+0.7858\right)\right]$$
where h is the relative humidity of the air (%), $q_{D}$ is the surface WT (℃), and $e_{a}$ is the saturated vapor pressure, (hPa).

Based on the process of nutrients–phytoplankton–zooplankton (N–P–Z), the CAEDYM can be used as a sub-model, coupled with the DYRESM, to simulate the dynamic process of the lake’s carbon, nitrogen, and phosphorus cycles, inorganic suspended particles, and algal extinction; this can accurately predict the overall changes in water quality. CAEDYM has the function of simulating the respiration, death, and metabolism of 7 species of phytoplankton, 5 species of zooplankton, and fish and jellyfish, along with their interactions with the external environment. As cyanobacteria and diatom are dominant species at Lake Erhai [32], so we mainly considered the growth and death for the two species. The key parameters of CAEDYM are listed in Table 3.

#### 3.1.2. DYRESM–Based Water Balancing Method (DWBM)

Lake volume is influenced by inflows, outflows, groundwater, evaporation and water supplies for drinking, agriculture and industry. Most of these parameters affecting water balance of lake were partly measured and it was very difficult to get satisfied inflow data required by model. So, we used DYRESM to calculate the inflows as accurately as possible based on the measured inflows, the measured outflows, the water supplies for drinking/agriculture/industry, the measured WLs and the relationship between WLs and their corresponding water storage capacities. The specific steps are as follows: (1) Establish the curve between the WLs and their corresponding water storage capacities by regression method for Lake Erhai. (2) Run DYRESM to get the daily WLs from model outputs. (3) Compare the simulated daily WLs with the measurements to calculate the unbalanced water volumes (UWV). (4) If the modeled WL is greater than the measure WL at the same day, the UWV will be added to the outflows. Otherwise, it will be added to the inflows for compensation. DWBM has been successfully applied to Lake Dianchi [33] and Lake Fuxian [34].

#### 3.1.3. Calibration and Validation

The monthly water quality, the WTs of surface layer (0.5 m depth) and the WLs were used to calibrate and validate DYRESM–CAEDYM. The model was calibrated based on the data for 2009–2012 and validated with the measurements in 2013 because the above parameters during 2009–2013 were relatively more continuously measured so the data were more reliable compared to other years. Meanwhile the water quality during this period was more stable, which can favor good model performance. The comparisons between simulations and measurements for WL, WT, and water quality are shown in Figure 2.

To gain further insight into the performance of model, Pearson’s correlation coefficients between the model outputs and monitoring data were calculated for all the variables. The *r* values of WL, WT, Chla, NH_4_-N, TN, and TP in calibration were 0.99, 0.96, 0.70, 0.15, 0.44, and 0.37 (*N* = 48) respectively. The *r* values in verification were 0.99, 0.98, 0.48, 0.41, 0.78, and 0.64 (*N* = 12) respectively. The results show that the model could reproduce WT, WL, TN, TP and Chla concentrations very well but had less successful performance for NH_4_-N (*r* = 0.15 calibration and *r* = 0.41 validation). 

#### 3.1.4. Estimation of External Loadings

The riverine TN and TP loads were calculated using the following formula:(2)$$W_{n}=\sum_{i=12}^{i}C_{i}\times Q_{i}$$
where $W_{n}$ represents the TN and TP loads entering the lake in the *n*th year (t/a), $C_{i}$ represents the mean concentrations of TN and TP in the river for the *i*th month (mg/L), and $Q_{i}$ is the total amount of water volume entering the lake in the *i*th month (m^3^/s).

### 3.2. Data Source

Detailed data information for hydrology, water quality, meteorology, and topography for Lake Erhai are shown in Table 4.

## 4. Results

### 4.1. Estimation of External Loads

The daily inflows estimated by DWBM and the collected water quality data for inflow rivers were used to calculate the annual input of external pollution loads. The estimated external loads were then compared to relevant data from literatures [21,24,25,39,40]. The results are shown in Figure 3. From 2000 to 2003, 2008, and 2019, the calculated values were significantly different from those in the literatures due to the over-estimated inflow by model. Pearson’s correlation analysis was carried out between the calculated TN and TP loads and the reported values from literatures. The results show that the correlation coefficients for TN and TP were 0.81 (*N* = 20, no literature data in 2006) and 0.86 (*N* = 21), respectively, indicating that the estimated pollution loads were reliable.

The average loads of TN and TP from 2000 to 2020 was 815.49 t/a and 55.85 t/a, respectively. The TN pollution load in 2002 was the highest with a value of 1891.34 t. It was the lowest in 2006 with a load of 107.31 t. The highest TP pollution load was 131.7 t in 2003 and the lowest load was 7.42 t in 2001.

### 4.2. Interannual Variation of External Loadings and Its Relationship with Rainfall

Figure 4 shows the annual rainfalls for 2000–2020, showing a strong correlation with both annual TN and annual TP loads. The rainfalls in 2000, 2001, 2002, 2008, and 2020 were greater than 1200 mm with relatively higher annual TN load (727.88 t/y) and TP load (46.8 t/y) for these years. The rainfalls were less than 800 mm in 2006, 2011, 2013, and 2014. The average annual TN and TP loads were 534.67 t/y and 37.29 t/y respectively for these years. So, less rainfall has caused less riverine TN and TP loads and lower water levels. 

### 4.3. Scenario

The government departments will take 2020 as the current base year by adopting the strategy of “space control, water conservation, source control, emission reduction, restoration and capacity, comprehensive management”. It was expected that the water quality of 27 main inflow rivers would all meet Grade Ⅲ or higher standard. The lake water quality would be at Grade Ⅱ in 2025. According to the calculations, the TN and TP loads in 2020 were 515.42 t and 37.34 t respectively, which were used to feed the model as the baseline. The six scenarios were set up based on the baseline (Table 5).

In the baseline scenario, no external loadings were reduced. The external pollution loads in scenarios 1–5 was reduced by 10%, 20%, 30%, 40%, and 50%, respectively. The sixth scenario was based on the assumption of good water quality in the all-inflow rivers with water quality meeting Class Ⅲ (TN = 1 mg/L, TP = 0.2 mg/L). The annual external TN and TP loads were 231.64 t and 27.75 t respectively, which were equivalent to 53.71% and 97% of the loads in the baseline scenario.

In the all scenarios, the model had the same initial WL (1984.6 m), and the same initial lake water quality with TN of 0.49 mg/L, TP of 0.02 mg/L and Chla of 7.9 μg/L. The simulation period was from January 1, 2020 to December 31, 2025. The external loads kept the same for 2020–2022 with the baseline scenario and they started to reduce from January 1, 2023 to December 31, 2025. The model outputs in 2025 were then used for analysis. The time step for model was 1 hour.

### 4.4. Water Quality Responses to External Loading Reductions

Figure 5 shows the comparisons of modelled TN, TP, and Chla for the above six scenarios and the baseline. In the baseline scenario, the TN concentration will be higher than 0.5 mg/L (Grade II) for March–October 2025 while there is no any pollution controls. The highest TN was 0.57 mg/L in early September. TP shows an obvious seasonal variation with peak values in summer (Figure 5b), which might be caused by significant input of external loads in rain season. Without reduction of external loads, TP concentration would be higher than 0.025 mg/L in late July, which could not meet Grade II standard. Chla varied with TN and TP with the highest value of around 40 μg/L in September, showing high risk of algal bloom at this time.

Water quality in 2025 was significantly improved after reducing the external pollution loadings during 2023–2024. Higher reduction intensity caused lower concentrations of TN, TP and Chla. TN show the best improvement effect when the pollution load was reduced by 20% or more. The peak value of TN dropped below 0.5mg/L with water quality meeting Grade Ⅱ throughout the year for S2–S6. The TP concentrations can also meet Grade Ⅱ except for all the scenarios with peak values occurring in summer. The concentrations of TN and TP decreased greatly when the pollution loads were reduced by more than 30%.

Chla decreased with increased reduction of TN and TP loads. Its peak values appeared in autumn following the same pattern with TN for all the scenarios and baseline. Chla can greatly decreased only when the external loads were reduced by more than 40% (i.e., S4, S5, S6). 

### 4.5. Comparison of Simulation Results

Reduction of external loadings could significantly reduce the concentrations of TN, TP and Chla in the lake. The improving degree of lake water quality was proportional to the reduction rate of external loadings (Table 6). In the scenario 1 with external loadings reduced by 10%, the water quality met the expected Grade II in seven months through the year, but the annual average Chla concentration was even higher than that in the baseline. After the reductions of external loading by 20% and 30%, the TN and TP met Grade II, but the Chla was only improved by 11.75% and 9.48% respectively. The Chla concentration was significantly reduced only if the external loading reduction rate was 40% or more, which suggests that there might be high risk of algal blooms without the controls on non-point source from its catchment at Lake Erhai.

In scenario 6, the concentrations of TN, TP, and Chla were 0.27 mg/L, 0.013 mg/L, and 4.76 μg/L, with an improvement of 46.32%, 38.74%, and 37.07% respectively compared to TN, TP, and Chla in 2020. So, the lake water quality can meet Grade II when the water quality of inflow rivers meets Grade Ⅲ.

## 5. Discussion

The *r* values in Figure 2 show good agreement between the simulations and the observations for surface WT and WL. The model also successfully reproduced the surface nutrient concentrations during both the calibration and validation periods. The simulation results show that reducing the loads by 10% had little effect on improving the TN and TP concentrations, which agrees with the conclusions by Yu et al. [18]. However, in contrast to their conclusions, when we reduced the load by 20%, the TN and TP can be greatly reduced. This might be caused by reductions of both TN and TP at the rate in our paper, or resulted from different models.

There were some discrepancies between the model outputs and the field measurements. Particularly, there was a poor fit between the modeled and observed data for TP. This might be caused by the over- or under- estimations of inflow volumes, or unsuccessful simulation of internal release of TP from sediment under anaerobic condition. In most of water quality models, TP is more difficult to capture than TN due to its more complex cycling. There are less parameters for TP than those for TN in DYRESM–CAEDYM, bring more difficulty to accurately predicate TP than TN [41]. Therefore, to provide more precise prediction of TP, more work might be conducted to know the actual phosphorus cycling in a specific lake, which should be incorporated into a model. 

Our model generally underestimated Chla in winter. The reason for this is that the model only included cyanophytes and diatoms, which account for approximately 60–80% of the phytoplankton biomass in the lake. Horizontal heterogeneity might be another fundamental source of this error. The validity of the one-dimensional hypothesis increases with depth. Therefore, it can reduce the impact of spatial changes on lake water quality [42]. Although a one-dimensional model will be difficult to capture spatial heterogeneity of water quality, phytoplankton, and zooplankton in lakes, it is still a useful management tool for decision-making and technical support.

The external loads of inflow rivers influence the lake water quality to a large extent [24]. From the monitored data of 23 main rivers, the Water Environment Monitoring Center of Dali found that the river-discharged loads at the north of lake accounted for 47.0%, 28.7%, and 61.9% for TN, TP, and COD respectively [36]. This may be result from the large number of livestock with domestic animal farmings in that area. Therefore, reducing domestic animals at the north of lake must be favoring eutrophication mitigation [43].

Water quality is normally nonlinearly responses to external loading reductions [44], which was proved in our simulations. With decreased pollution loads, the lake nutrient concentrations gradually decrease but not linearly proportionally. For example, with reduction rate of 10% for external TN and TP loads, the lake TN and TP decreased limitedly and the lake Chla slightly increased. However, the lake TN, TP and Chla decreased remarkably with a reduction rate of 20%, indicting there is obvious nonlinearity between external loading reduction and lake water quality improvement. This nonlinearity requires model and data mining technologies to be used for better water environment management.

Chla variations followed the same seasonal pattern with TN with peak values in late summer or early autumn, while TP peak values were found in middle summer. This indicates that the algal growth is nitrogen-limited, which agrees with Wu.et al. [45]. This also indicates that the lake TN might be mainly controlled by external loads from catchment but the lake TP is probably regulated by both external loads and internal release with stratification in middle summer because the anaerobic bottom water with low oxygen generated by stratification can favor phosphate release. Xiao et al [24] reported the internal TN and TP loads accounted for 31.99% and 14.23% of the total inputs from point and non-point sources respectively from 2008 to 2018. So, mitigation of eutrophication will require concerted efforts to reduce long-term nutrient inputs and possibly necessitate addressing internal loads at Lake Erhai.

## 6. Conclusions

(1) DYRESM–CAEDYM successfully reproduced the concentrations of TN, TP and Chla at Lake Erhai, showing its strong capability and reliability in this lake although it had less success in predicting TP than TN and Chla.

(2) With more than external loading reduction rate of 20%, the lake TN and TP can meet expected Grade II standard of water quality. 

(3) Phytoplankton is nitrogen-limited from our model outputs and there might be still high risk of seasonal algal blooms with external loading reduction. Internal release is suggested to be seriously considered for mitigating the eutrophication at Lake Erhai.

## Figures and Tables

**Figure 1 ijerph-20-05038-f001:**
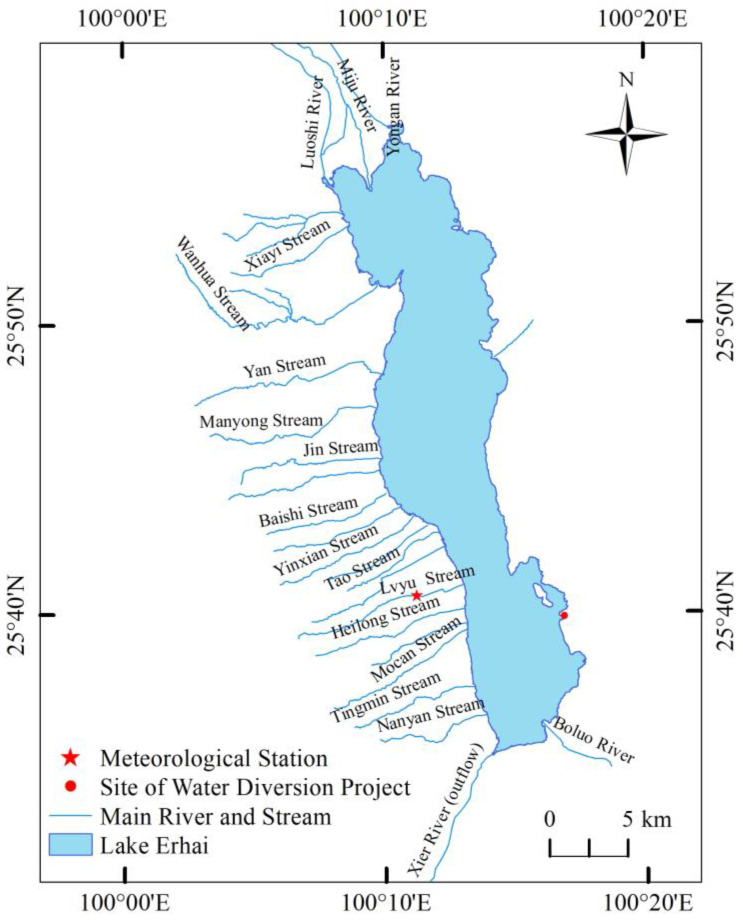
Lake Erhai and its main rivers and streams.

**Figure 2 ijerph-20-05038-f002:**
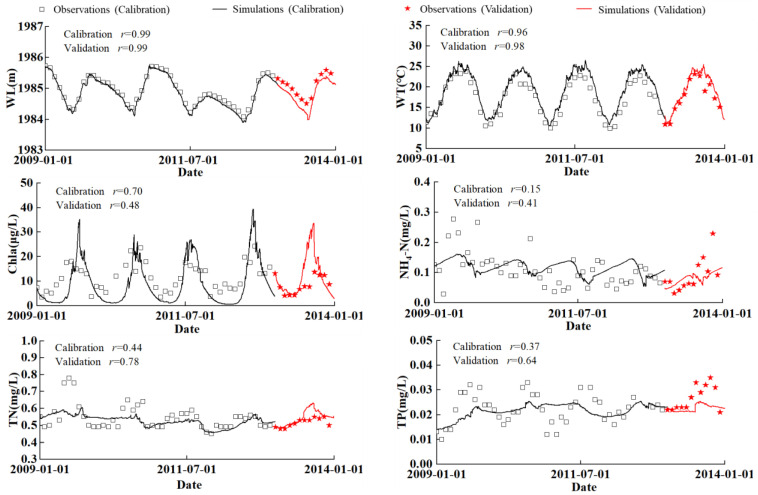
Comparisons between simulations and measurements for WL, surface WT, and water quality parameters (TN, TP, Chla, NH_4_-N) during calibration (2009–2012) and validation (2013).

**Figure 3 ijerph-20-05038-f003:**
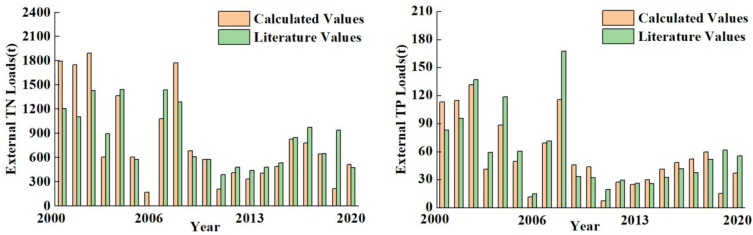
Comparison of the estimated TN and TP loads (orange) and the reported TN and TP loads from literatures (green) at Lake Erhai for 2000–2020.

**Figure 4 ijerph-20-05038-f004:**
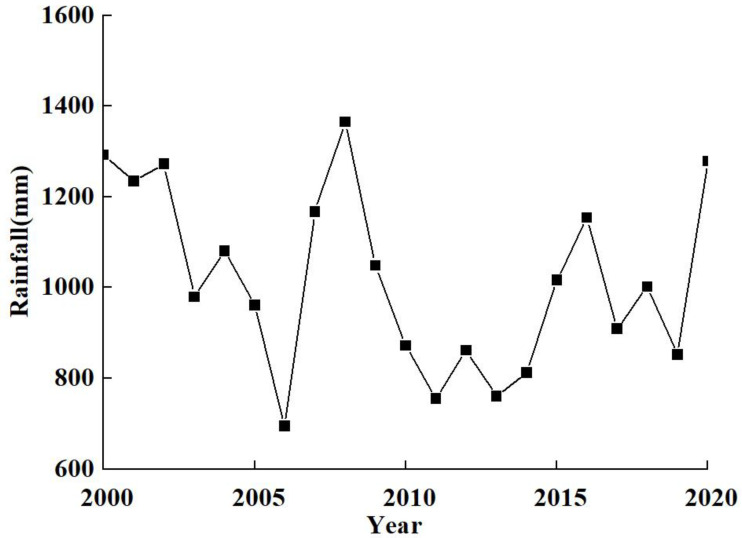
Annual rainfalls for 2000–2020 at Lake Erhai.

**Figure 5 ijerph-20-05038-f005:**
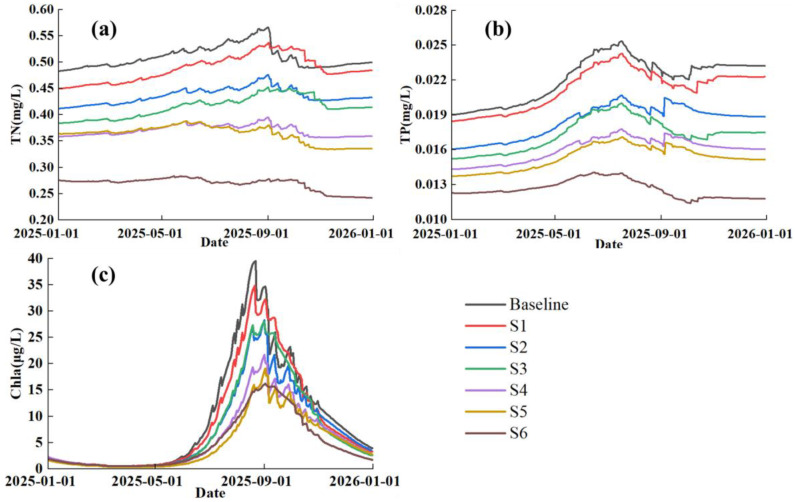
Simulated daily TN (**a**), TP (**b**), and Chla (**c**) under the baseline and six scenarios with different reduction rates of external TN and TP loads at Lake Erhai in 2025.

**Table 1 ijerph-20-05038-t001:** The environmental quality standards for surface water of China (GB3838-2002).

Grade	Surface Water	I	II	III	IV	V
TN (mg/L)≤	Lake, River	0.2	0.5	1.0	1.5	2.0
TP (mg/L)≤	River	0.02	0.1	0.2	0.3	0.4
	Lake	0.01	0.025	0.05	0.1	0.2

**Table 2 ijerph-20-05038-t002:** Main parameters for DYRESM.

Parameters	Reference Value	Value in This Paper
Bulk aerodynamic momentum transport coefficient	1.3 × 10^−3^~2.17 × 10^−3^ [27,28]	1.3 × 10^−3^
Mean albedo of water	0.07~0.08 [28,29]	0.08
Emissivity of a water surface	0.94~0.96 [16,27]	0.96
Critical wind speed	3.0~6.5 m/s [16,27]	4.5 m/s
Shear production efficiency	0.06~0.08 [16,27]	0.08
Potential energy mixing efficiency	0.2~0.29 [16,27]	0.25
Wind stirring efficiency	0.15~0.8 [26,30]	0.25
Vertical mixing coefficient	200~400 [27,31]	400

**Table 3 ijerph-20-05038-t003:** Main parameters in CAEDYM.

Parameter	Value
Nitrification rate coefficient	0.006/day
Denitrification rate coefficient	0.05/day
PO_4_ sediment flux	0.85 g/m^2^/day
NH_4_-N sediment flux	0.01 g/m^2^/day
NO_3_-N sediment flux	−0.035 g/m^2^/day
Maximum potential growth rate of cyanobacteria	0.4/day
Respiration rate coefficient for cyanobacteria	0.04/day
Maximum potential growth rate of diatom	0.3/day
Respiration rate coefficient for diatom	0.06/day
Half-saturation constant for phosphorus	0.09 mg P/L
Half-saturation constant for nitrogen	0.05 mg N/L

**Table 4 ijerph-20-05038-t004:** Data information.

Data Type	Period	Description	Data Source
Lake topography	—	Water depth	Erhai Administration Bureau
Meteorology	2000~2020	Daily meteorological data from Dali Meteorological Station (NO.56751, 25.7° N, 100°18′41′′ E)	The China Meteorological Data Service Center
outflows	2000~2020	Annual data were distributed to each month of the year according to the proportion of rainfall, and then evenly distributed to each day of the month	Wu et al., 2020 [35]
Water quality of inflow rivers^1^	2008~2019	Monthly TN and TP, missed values were replaced by multiyear averaged value in the same month	Yu et al., 2011; Zhao et al., 2012; Yan et al., 2020; Huang et al., 2016; [18,36,37,38]
WLs	2000~2020	Daily WLs	Erhai Administration Bureau
Vertical profile of water quality ^1^	January~December 2015	Monthly data, WT, and DO data were collected at 1 m interval	Zhu et al., 2017 [22]
Water quality of Lake Erhai ^1^	2008~2020	Monthly monitoring by conventional method	Erhai Administration Bureau

^1^ TN was determined by ultraviolet spectrophotometry with potassium persulfate digestion. TP was determined by ammonium molybdate spectrophotometry.

**Table 5 ijerph-20-05038-t005:** Six scenarios with different reduction rate of external loads.

Scenarios	Reduction Rate of External Loads	TN Loads (t/y)	TP Loads (t/y)
Baseline	0	515.42	37.34
S1	10%	463.88	33.61
S2	20%	412.34	29.87
S3	30%	360.79	26.14
S4	40%	309.25	22.40
S5	50%	257.71	18.67
S6	Grade Ⅲ for water quality of inflow rivers	231.64	27.75

**Table 6 ijerph-20-05038-t006:** Concentrations and improvement rates of TN, TP and Chla for the baseline and the six scenarios at Lake Erhai.

Scenarios	TN		TP		Chla	
	Annual Concentration (mg/L)	Improvement Rate (%)	Annual Concentration (mg/L)	Improvement Rate (%)	Annual Concentration (mg/L)	Improvement Rate (%)
Baseline	0.51	—	0.021	—	8.30	—
S1	0.49	3.27	0.020	4.15	8.72	−4.37
S2	0.44	13.42	0.019	12.32	7.20	11.75
S3	0.41	17.57	0.017	17.46	7.39	9.48
S4	0.37	26.37	0.016	23.28	5.66	27.78
S5	0.36	27.75	0.015	26.43	4.81	36.63
S6	0.27	46.32	0.013	38.74	4.76	37.07

## Data Availability

The data that support the findings of this study are available from the corresponding author upon reasonable request.

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
