# Peer review of "Quantitative Assessment of Water Quality Improvement by Reducing External Loadings at Lake Erhai, Southwest China"

_ijerph, 2023, doi:10.3390/ijerph20065038_

Round 1

Reviewer 1 Report

Line 146 needs clarification: this paper summarized all river channels 146 entering and leaving Lake Erhai into one total river channel and one total river channel 147 leaving the lake. 

Why the calibration period is not recent? It is for 2009-2012

The water quality, which is very important in this research is only refered to a Chinese paper, how the authors are sure about the quality of data? Even the methods of measurement is not mentioned, either the quality control.

How the authors justify the results of this model comparing to other models?

Reviewer 2 Report

Dear Authors, this paper discusses the very important problem of eutrophication of Erhai Lake, which is of serious concern in relation to the maintenance of the drinking water quality standard.

The structure and logical order of the presented topics in the manuscript has been maintained. There are some comments below on how this article could be improved according to some details on the presentation of simulation results and added practical value in conclusions. Please see my suggestions:

1. In the introduction to the manuscript, I propose to present the existing water quality standards according to GB3838-2002 (e.g. in tabular form) and refer to the quality of these waters in other states.

2. Is there any observations or assumptions about which areas of the Lake Erhai may be most affected by exogenous pollutants and may have an impact on the deterioration of water quality for the entire water reservoir? Which rivers or streams entering into the lake may be significant contributors to pollution load ?

3. With reference to the elaboration of the results, I propose to make a summary table, which will indicate how much more the effectiveness of the reduction of individual parameters TN, TP, Chla should be increased in order to achieve the legally required water standard. This will facilitate the analysis and readability of simulation results.

4. What solutions are assumed to ensure the continuity of drinking water supply at the legally required standard, during the period when the standards of exogenous compounds are not met?

5. Please indicate the strengths and weaknesses of the simulation performed. Did the analysis provided with a satisfactory answer regarding the quality outlook of Erhai Lake? What would need to be improved and included in the analyses conducted in order to study the problem more thoroughly?

6 What further measures would need to be taken to reduce the deteriorating water quality of the lake in the long term?

Sincerely, Reviewer

Reviewer 3 Report

The authors evaluated the impact of exogenous cut-back in river channels on the water quality of Lake Erhai using a one-dimensional hydrodynamic and ecological coupling model (DYRESM-CAEDYM) that accounts for both water level and water quality.  Six different scenarios were considered, each with different intensities of load reduction.

The study deals with an interesting issue and is globally well articulated. However, the authors have not sufficiently emphasized the aspects of novelty and originality.

Reading the text was often challenging because of its form, which severely penalizes the value of the content.

The English language used in the text needs significant improvement to be clearer and more concise. The text is full of grammatical and punctuation errors, and sentences are often long and disjointed, making it difficult to read, especially in the abstract which needs to be entirely rewritten.

Many sentences (e.g., lines 19-22, 30-32, 55-58, 71-74, 132, etc.) do not make much sense and must be rephrased.

Based on my assessment, the paper is not ready for publication in its current state.

Reviewer 4 Report

The research article "Quantitative assessment of water quality improvement by exogenous reduction at Lake Erhai of Southwest China", could be considered for publication in this prestigious journal after some minor modifications. 

Comment # 01 The Abstract should be shorten to include only the basic points of the study. Extensive details should not be discussed here. 

Comment # 02 Extensive English language check is required in terms of grammar and typo errors. 

For instance:

In the first line of the abstract, either use improvement or effect, "improvement effect" does not make any sense. 

In line 18 and 19, information of six set of scenarios is a repeated information.

In line 20, "Total Nitrogen (TN) and Total Phosphorous (TP) decreased were both reduced by.....", decreased and reduced are the same thing. use either one of them.

In the similar line, the verb of "were" was used twice. The particular phrase "were compared with no reduction (base line)" should be replaced with "as compared to no reduction (base line)". 

Long complex sentences, having three or four phases should be avoided and shorter sentence should be used.

The line 32 page 1 "water quality has not improved significantly when exogenous reduction of 10%", the "when" in the sentence should be replaced by "with the".

The line 40 page 1, "For the purpose of improving the water quality, we must pay attention to the control endogenous pollution",  the sentence should be improved to "the control of endogenous pollution".

In page 2 line 59, please correct the form of verb by replacing the "measure" with "measures".

In page 2 line 69, please replace the word "high frequency" with "high-frequency".

In page 2 line 72, please remove the comma after the word that.

In page 2 line 95, add the space between the last two lines.

In page 3 line 123, replace the word "lake" in "along the rivers and lake in the north...." by lakes.

In page 3 line 118, replace the word "the" in "with the annual average" with "an".

In page 4 line 146, replace the word of "of" in "the absence of monitoring data of river flow" with "the absence of monitoring data on river flow".

In page 5 line 156, correct the line "Meteorology is an important driving factor of lake water quantity...." to "Meteorology is an important driving factor in determining lake water quantity".

In page 9 line 282, correct the typographical error as the preposition of "in" has been repeated twice by mistake.

In page 9 line 286, replace the word "of" in line "The massive increase of exogenous pollution" by the word "in". 

In page 9 line 290, correct the line "the highest values is about..." to "the highest value is about..".

In page 9 line 301, correct the line "when reduce 30% and 40%" to "when reduce by 30% and 40%".

In page 10 line 313, correct the line "reached the standard is only seven months, and" to "reached the standard in only seven months and".

In page 11 line 332, correct the line "the average TN and TP load were" to "the average TN and TP loads were"

In page 11 line 337, correct the line "with the increase of pollution..." to "with the increase in pollution".

In page 12 line 372, correct the line "no effect on winter and spring" to "no effect in winter and spring". 

Comment # 03 The abbreviations used in the abstract should be repeated once again in the manuscript text as well when used for the first time.

Comment # 04 The scientific language should be used and words like obviously (page 4 line 152) and really (page 9 line 284) should be removed from the manuscript. 

Comment # 05 In the table 03, replace the word references by proper way of references such as "Li et al. "

Comment # 06 The comparative analysis of conventional methodologies vs simulation methodologies for evaluating water quality of the lakes should also be discussed in the introduction section. 

Comment # 07 The ending paragraph of the introduction section should include the overview of the whole methodology.

Comment # 08 The table 1 should include all the relevant units associated with the main parameters in DYRESM model.

Comment # 09 Provide the justification of selecting only these main parameters for modelling. Plus also mention any parameter ignored in the DYRESM model in the materials and method section as well.

Comment # 10 Similar comment # 09 is applied for the CAEDYM model as well.

Comment # 11 In section 3.1.3, authors suggested that elected model exhibit significant variability and reproduction. Support this significant argument with reference. Furthermore, also provide the statistical justification of the validity of the model.

Comment # 12 Provide the reasoning behind the observed vast differences in the validation period and calibration period with respect to different parameters (presented in figure 2).

Comment # 13 Provide a separate section discussing the Limitations of the selected model in the results and discussion section.

Round 2

Reviewer 2 Report

The manuscript has been improved by the Authors. There are no additional comments.

Author Response

Thanks for your comments
